# ICI 182,780 Attenuates Selective Upregulation of Uterine Artery Cystathionine β-Synthase Expression in Rat Pregnancy

**DOI:** 10.3390/ijms241814384

**Published:** 2023-09-21

**Authors:** Jin Bai, Yao Li, Guofeng Yan, Jing Zhou, Alejandra Garcia Salmeron, Olamide Tolulope Fategbe, Sathish Kumar, Xuejin Chen, Dong-Bao Chen

**Affiliations:** 1Department of Obstetrics and Gynecology, University of California Irvine, Irvine, CA 92697, USA; baij3@hs.uci.edu (J.B.); garcias9@uci.edu (A.G.S.); fategbeo@uci.edu (O.T.F.); 2Department of Laboratory Animal Sciences, School of Medicine, Shanghai Jiaotong University, 280 South Chongqing Road, Shanghai 200025, China; yao.li@shsmu.edu.cn (Y.L.); 181864@shsmu.edu.cn (G.Y.); 183008@shsmu.edu.cn (J.Z.); chenxuejin@cuhk.edu.cn (X.C.); 3Department of Comparative Biosciences, University of Wisconsin-Madison, Madison, WI 53706, USA; sathish.kumar@wisc.edu

**Keywords:** uterine artery, cystathionine β-synthase, endogenous estrogens, estrogen receptors, pregnancy

## Abstract

Endogenous hydrogen sulfide (H_2_S) produced by cystathionine β-synthase (CBS) and cystathionine-γ lyase (CSE) has emerged as a novel uterine vasodilator contributing to pregnancy-associated increases in uterine blood flow, which safeguard pregnancy health. Uterine artery (UA) H_2_S production is stimulated via exogenous estrogen replacement and is associated with elevated endogenous estrogens during pregnancy through the selective upregulation of CBS without altering CSE. However, how endogenous estrogens regulate uterine artery CBS expression in pregnancy is unknown. This study was conducted to test a hypothesis that endogenous estrogens selectively stimulate UA CBS expression via specific estrogen receptors (ER). Treatment with E_2_β (0.01 to 100 nM) stimulated CBS but not CSE mRNA in organ cultures of fresh UA rings from both NP and P (gestational day 20, GD20) rats, with greater responses to all doses of E_2_β tested in P vs. NP UA. ER antagonist ICI 182,780 (ICI, 1 µM) completely attenuated E_2_β-stimulated CBS mRNA in both NP and P rat UA. Subcutaneous injection with ICI 182,780 (0.3 mg/rat) of GD19 P rats for 24 h significantly inhibited UA CBS but not mRNA expression, consistent with reduced endothelial and smooth muscle cell CBS (but not CSE) protein. ICI did not alter mesenteric and renal artery CBS and CSE mRNA. In addition, ICI decreased endothelial nitric oxide synthase mRNA in UA but not in mesenteric or renal arteries. Thus, pregnancy-augmented UA CBS/H_2_S production is mediated by the actions of endogenous estrogens via specific ER in pregnant rats.

## 1. Introduction

During pregnancy, the organ systems throughout the mother’s body make adaptive changes to optimize the uterine environment to safeguard pregnancy health, with the most dramatic changes in the cardiovascular system [1,2]. Maternal vascular adaptations to pregnancy result in gestation-age-dependent increases in uterine blood flow (UtBF) up to 20–50-fold. This is mandatory for delivering maternal nutrients and oxygen to the fetus and for exhausting respiratory gases and wastes from the fetus [3,4]. Insufficient rises in UtBF during pregnancy lead to placental ischemia/hypoxia, further resulting in placental under-perfusion, representing a major pathophysiology underlying many pregnancy-specific disorders such as preeclampsia, fetal growth restriction (FGR), and preterm birth [5,6,7]. These diseases not only deteriorate maternal and fetal wellbeing during pregnancy but also derail the lifelong health trajectories of the mother and her child [8].

Concurrently, endogenous estrogen production is significantly elevated in pregnancy [9] and levels of total plasma estrogens can reach as high as ~1000-fold in pregnant vs. nonpregnant women [10]. Utilizing intact and ovariectomized sheep models, studies have shown that a marked rise in UtBF occurs as early as 15–30 min after a bolus subcutaneous injection of exogenous estradiol-17 β (E_2_β, 1 µg/kg body weight), reaching its maximum at 90–120 min and decreasing thereafter; however, it remains elevated up to 7–10 days [11,12,13]. Although the importance of estrogens in pregnancy is well recognized [14], the mechanisms underlying estrogen-induced uterine vasodilation remain partially understood. However, a large body of evidence accumulated since the 1990s favors a leading role of nitric oxide (NO) in the mechanism. This is locally produced by the uterine artery (UA) endothelium via upregulating the expression [15,16,17] and activation [17,18] of endothelial NO synthase (eNOS). Systemic E_2_β administration and local inhibition of either the actions of NO synthase with L-N^G^-Nitro arginine methyl ester (L-NAME) [13,19,20] or estrogen receptors (ERs) with ICI 182,780 (ICI) [13] inhibit up to ~68% of estrogen-induced uterine vasodilation. In vivo studies have established that L-NAME and ICI partially reduce (25–30%) UtBF from its maximum levels during pregnancy [13]. These studies delineate the cause–effect relationships among endogenous E_2_β and de novo synthesis of NO through eNOS and ERs while also implicating mechanisms in addition to eNOS-NO to mediate uterine hemodynamics. While preclinical studies based on theories regarding NO-mediated mechanisms in upregulating uterine–placental perfusion have succeeded in various animal models of preeclampsia [21], clinical trials targeting NO pathways have thus far achieved little to no success in these diseases [22,23], requiring more studies to identify other mechanisms.

The proangiogenic vasodilator hydrogen sulfide (H_2_S) is endogenously produced mainly from L-cysteine by cystathionine β-synthase (CBS) and cystathionine γ-lyase (CSE) [24,25]. We have shown that through selective upregulation of endothelial cell (EC) and smooth muscle cell (SMC) CBS expression [26,27,28], UA H_2_S production is stimulated by exogenous E_2_β treatment in ovariectomized sheep [26] and also positively correlates with endogenous estrogens in sheep [29] and women [28]. H_2_S stimulates pregnancy-dependent relaxation of pressurized UA rings ex vivo [28,30] via activating SMC large conductance Ca^2+^-activated voltage-dependent potassium channels [30], which mediate estrogen-induced UA dilation in pregnancy [31,32]. Thus, H_2_S is a novel UA dilator alongside NO to mediate uterine hemodynamics.

Utilizing primary ovine UA endothelial cell (UAEC) and UASMC models, we reported that E_2_β stimulates primary ovine UAEC and UASMC H_2_S production in vitro by stimulating specific ER-dependent upregulation of *CBS* transcription involving ERα and ERβ [27,33]. E_2_β also stimulates CSE expression in these ovine UA cell models, which is contrary to in vivo conditions [26,28,29]. However, in primary human UA EC, E_2_β stimulates UAEC H_2_S production by stimulating specific ER-dependent *CBS* transcription via direct ERα and ERβ interactions with the proximal *CBS* promoter estrogen-responsive elements (EREs) [34], showing species-dependent ER-mediated mechanisms controlling UA CBS/H_2_S production in vitro. In vivo, UA CBS/H_2_S production is augmented in the two physiological states of elevated endogenous estrogens [35]: in the proliferative/follicular phase of the ovarian cycle and pregnancy in women [28] and sheep [29]. Although these studies implicate the role of endogenous estrogens, the process by which endogenous estrogens regulate UA CBS expression in pregnancy is currently elusive. Pregnant animals receiving ICI have been used to ascertain the role of endogenous estrogens in gene expression [36] and pregnancy-associated uterine vasodilation [13]. Therefore, we conducted this study using pregnant rats treated with ICI to test a hypothesis that endogenous estrogens stimulate UA CBS expression via specific ER in vivo.

## 2. Results

### 2.1. E2β Stimulates CBS Expression via ER Mediation in Rat UA Ex Vivo

Baseline CBS mRNA was numerically higher in P vs. NP UA, but the difference was not statistically significant. Treatment with E2β significantly stimulated CBS mRNA in a concentration-dependent manner in organ cultures of NP and P rat UA rings in vitro. Treatment with E2β (1 and 10 nM) for 24 h significantly stimulated CBS mRNA expression in both NP and P rat UAs. The stimulatory effects of E2β on CBS mRNA further increased with 100 nM E2β, reaching its maximum level in NP UA by 4.05 ± 0.51-fold vs. control (*p* < 0.01) and in P UA by 5.35 ± 0.54-fold vs. control (*p* < 0.01), which were completely abrogated by 1 µM ICI (Figure 1A). In addition, the stimulatory effects of E2β on CBS mRNA in P UA were statistically greater at all tested concentrations (0.01–100 nM) of E2β than in NP UA (*p* < 0.01). Baseline levels of CSE mRNA did not differ in NP vs. P UA. Treatment with E2β (0.01–10 nM) did not alter CSE mRNA in both NP and P UA, but at 100 nM E2β also increased CSE mRNA expression with similar potency in NP and P rat UA, which was blocked using ICI (Figure 1B).

### 2.2. ICI Decreases UA but Not Systemic Artery CBS mRNA in Rat Pregnancy In Vivo

In comparison to vehicle-treated P rats, systemic administration of ICI for 24 h significantly decreased GD20 P rat UA CBS mRNA by 62 ± 6% (*p* < 0.01, n = 8) without altering levels of other systemic mesentery artery (MA) and renal artery (RA) CBS mRNAs. ICI treatment also decreased UA eNOS mRNA by 51 ± 21% (*p* < 0.05, n = 9). ICI treatment did not change UA, MA and RA CSE mRNA, nor MA and RA eNOS mRNA in GD20 P rats in vivo (Figure 2).

### 2.3. ICI Decreases Rat UA Endothelial and SM CBS Protein

CBS and CSE proteins were immunolocalized in both EC and SMC of GD20 P rat UAs. Following systemic ICI treatment for 24 h, levels of CBS protein were significantly reduced by 24 ± 4% (*p* < 0.05, n = 3) in EC and 55 ± 3% (*p* < 0.05, n = 3) in SMC in the animals. However, UA EC and SMC CSE protein were not significantly altered by the ICI treatment in GD20 P rats (Figure 3).

## 3. Discussion

The vasodilatory effect of estrogens was initially described in a classical study by Markee (1932), which showed that treatment with crude estrogen extracts results in the vasodilatation (hyperemia) of uterine endometrial tissue transplanted to the anterior chamber of the eye [37]. In early studies using ovariectomized nonpregnant sheep models, exogenous E_2_β administrated either locally in the uterine artery with a low dose (3 μg) or a higher systemic dose (1 μg/ kg body weight) will result in a maximal and remarkably predictable pattern of increase in UtBF; the response begins to rise around 20–30 min, then gradually increases and reaches its maximum value up to 10-fold baseline at 90–120 min, thereafter decreasing but remaining elevated up to 7–10 days [11,12,38,39,40,41,42]. Exogenous estrogens also stimulate vasodilation in various systemic arteries, but with maximum response in the uterus [42,43]. The direct estrogenic uterine vasodilatory effect is of significance in perinatal medicine because: (1) endogenous estrogen levels increase throughout human pregnancy [10]; (2) UtBF increases up to 20–50-fold in human pregnancy, which is the lifeline of fetal development and survival as it arguably provides the only nutrients/oxygen sources for fetal/placental development [3,4]; (3) estrogen production is reduced in pregnant women who develop preeclampsia [44]; and (4) aberrant estrogen metabolism due to catechol-O-methyltransferase deficiency results in preeclampsia-like symptoms in mice [45].

The mechanisms underlying estrogen-induced uterine vasodilation have been a long-lasting point of research because this research not only delineates the uterine hemodynamics important for maternal and fetal health [5,6,7] but also provides knowledge relevant to solving the puzzle of the cardiovascular protective effects of estrogens [46]. Early pharmacological studies have shown that de novo protein synthesis is required for estrogen-induced uterine vasodilation. This is because the unilateral infusion of cycloheximide significantly inhibits E_2_β-induced UtBF elevation during the 90 min infusion, while the contralateral E_2_β-induced UtBF is unaffected. This inhibition lasts for more than 30 min after the removal of the cycloheximide infusion [39]. In addition, various estrogens, including E_2_β, estrone, estriol, Premarin, raloxifene, and extremely high doses of the anti-estrogen *trans*-clomiphene [11,12,38,39,40,41,42,47,48], can all stimulate UtBF with a similar pattern and efficacy, suggesting an involvement of specific ER-mediated mechanisms. This idea was indirectly supported by a study in which Lineweaver–Burk plots were developed using the reciprocal of UtBF responses vs. the dose of E_2_β and catechol estrogens. Given that the *y*-axis intercepts of the two estrogens were the same, it was suggested that these estrogens bind to the same receptors but have different affinities and thus vasodilatory potency, as evidenced by the differences in the *x*-axis intercepts [49].

UtBF fluctuates regularly during the estrous cycle in animals, with a substantial increase followed by a decrease during the periovulatory period [50,51]. The follicular phase is a time when E_2_β is produced by the developing follicles, and UtBF reaches maximum levels while progesterone (P4) is virtually undetectable [51]. During pregnancy, UtBF is elevated when levels of both E_2_β and P4 are high [52,53]. Similar uterine hemodynamics occur in the menstrual cycle [54] and pregnancy [4] in women, with comparable changing patterns of E2/P4 levels. Because P4 alone does not stimulate UtBF [55], the follicular phase and pregnancy are viewed as two physiological states of elevated endogenous estrogens that upregulate UtBF [56]. Uterine artery endothelium and vascular smooth muscle express both ERα and ERβ, which are regulated by endogenous (follicular and pregnancy) and exogenous estrogens, suggesting that the uterine artery is a target site for fluctuating estrogen levels [57,58,59].

Two types of anti-estrogens have been used to dissect ER-mediated mechanisms. This includes type I anti-estrogens that are called selective estrogen receptor modulators (SERMs), which are analogs of tamoxifen, and type II anti-estrogens that are pure anti-estrogens such as ICI 164,384 and ICI 182,780 [60]. SERMs are non-steroidal compounds that bind both ERα and ERβ and produce weak estrogen agonist effects in certain tissues while producing estrogen antagonist effects in others [61]. ICI 182,780 is a selective steroidal estrogen antagonist that blocks estrogen action by competing for binding ERs in estrogen-responsive tissues [62]. Zoma et al. (2001) first showed that ICI 182,780 completely blocked elevated UtBF response to exogenous tibolone (a hormone-replacement therapy in postmenopausal women) in nonpregnant ovariectomized sheep [63]. Magness et al. (2005) demonstrated in follow-up studies that ICI 182,780 inhibits ~65% of the maximum levels of E2β-induced UtBF in nonpregnant ovariectomized ewes; it also effectively inhibits baseline UtBF responses in the physiological states of elevated endogenous estrogens, follicular phase of the estrus cycle, and pregnancy in sheep [13]. These studies established that estrogen-induced uterine vasodilation is mediated by specific ERs.

Since the early 1990s, a large body of evidence has further shown enhanced NO production locally by UA endothelium as the leading mechanism to mediate estrogen-induced uterine vasodilation [13,19,20]. Enhanced UA NO production is mediated by the increased expression [15,16,17] and activation [17,18] of eNOS, which is mainly present in the endothelium. UA endothelial and smooth muscle cells express both ERα and ERβ [57,58,59]. Endothelial eNOS expression via estrogen stimulation is mediated by ERα interaction with the proximal eNOS promoter EREs [64]. Conversely, eNOS activation by estrogens likely involves its release from caveolar domains on the plasma membrane [65] and ser^1177^ phosphorylation by extracellular signal-activated kinases and protein kinase B/Akt via nongenomic pathways mediated by ERα localized on the plasma membrane caveolae [18,66,67].

Nonetheless, blockade of the NO pathway by L-NAME inhibits ~68% E2β-induced and ~26% baseline pregnancy-associated UtBF responses in sheep, which are similar to those inhibited by ICI 182,780 [13]. This suggests that other mechanisms exist alongside NO to mediate uterine hemodynamics regulation. To this end, our recent studies have shown that enhanced UA production of H_2_S, the third member of the gasotransmitter family [68], seems to serve this role [28,30]. We reported that estrogen replacement treatment in ovariectomized nonpregnant sheep stimulates UA H_2_S production by selectively upregulating EC and SM CBS (but not CSE) expression [26]. In that study, we have also shown that exogenous E2β stimulates mesentery artery EC and SM CBS (but not CSE) expression and H_2_S production without altering carotid artery EC and SM CBS expression and H_2_S production [26], showing vascular bed-specific effects of exogenous E2β on H_2_S biosynthesis in vivo. In follow-up studies, we have reported that UA H_2_S production is augmented in the follicular phase sheep, proliferative phase women, and ovine and human pregnancy [28,29], in association with elevated endogenous estrogens [10,52,53]. In addition, our mechanistic studies using primary EC and SMC culture models have further demonstrated that estrogen-stimulation of UA H_2_S biosynthesis is mediated by specific ER-mediated upregulation of *CBS* transcription involving direct interactions of ERα/β with the proximal *CBS* promoter EREs [27,33,34].

In an organ culture model of freshly prepared P vs. NP rat UA rings in vitro, we have previously shown that E2β stimulates pregnancy-dependent type II angiotensin receptor (AT_2_R) expression associated with elevated endogenous estrogens in pregnant rats [59]. With this model, we show here that E2β stimulates UA CBS mRNA expression but with different potency in P vs. NP rat UA rings in vitro. As little as 0.01 nM E2β is effective in stimulating CBS mRNA in P vs. NP UA, and this pregnancy-dependent CBS mRNA upregulation is consistent in all E2β concentrations (0.01 to 100 nM) tested. E2β stimulates CBS mRNA in NP UA rings in vitro, but the effective concentrations are at 1–100 nM, which are higher than in P UA. ICI blocks E2β (100 nM)-stimulated CBS expression in both NP and P UAs. At 100 nM, E2β also stimulated CSE mRNA, but this stimulation is not pregnancy dependent. The findings differ from those in our in vivo studies showing UA CBS but not CSE upregulation via E2β replacement treatment and in pregnancy in vivo [26,28,29]. However, the findings agree with our previous studies using ovine UAEC and UASMC models in vitro [27,33]. In addition, baseline CBS expression is only numerically higher but does not reach statistical significance in rat P vs. NP UA, contrasting our previous studies in sheep [29] and women [28]. The cause of these discrepancies is unclear but likely originated from in vitro culture conditions and species-related effects. Nonetheless, our current study provides further evidence that exogenous E2β selectively stimulates CBS expression in rats and is mediated by specific ERs. Of note, 0.01-1 nM E2β are in the physiological range while 10-100 nM E2β are supraphysiological concentrations in women; however, for in vitro mechanistic studies, these concentrations may reflect the effects of total estrogens seen in pregnant women conceived after ovarian stimulation and in vitro fertilization [10].

Pregnant animals receiving ICI 182,780 have been previously used to address the role of ERs in endogenous [13] and exogenous [63] estrogen-induced uterine vasodilation and expression of uterine myometrial genes, including inducible NOS by endogenous estrogens [36]. Because the role of ERs in UA and systemic artery CBS/CSE expression by endogenous estrogens has never been tested, we therefore used pregnant rats receiving ICI 182,780 as a model to determine if systemic administration of ICI 182,780 (0.3 mg/rat subcutaneous injection) would affect UA and systemic (mesenteric and renal arteries) CBS and CSE expression in vivo. ICI treatment for 24 h significantly inhibits UA but not systemic MA and RA CBS without altering CSE mRNA levels in GD19 pregnant rats in vivo. In addition, immunohistochemical analyses have also shown that ICI treatment significantly reduces UA EC and SM CBS but not CSE protein in UA in pregnant rats in vivo. In addition, ICI reduces UA but not MA and RA artery eNOS mRNA levels in pregnant rats. As is consistent with numerous previous studies showing UA endothelial eNOS upregulation by estrogens and pregnancy [15,16,17,69,70], simultaneous inhibition of CBS and eNOS expression by ICI treatment in pregnant rats in vivo suggests that the CBS/H_2_S system, alongside eNOS/NO, plays a role in mediating estrogen-induced uterine vasodilation in pregnancy. 

Estrogens signal via both genomic and nongenomic pathways [71]. The former is mediated by ligated ERs (ERα and ERβ) that function as transcription factors to interact with promoter EREs to initiate target gene expression. ICI 182,780 was initially developed as a pure anti-estrogen with a high affinity to ERα and ERβ so that it blocks estrogen actions by competing for binding ERs in estrogen-responsive tissues [62]. Estrogens can also initiate rapid cellular responses via nongenomic pathways by interacting with membrane ERα/ERβ and the G protein-coupled receptor 30/G protein-coupled estrogen receptor 1 (GPR30/GPER1) [71]. Rat UA GPER1 expression increases during gestation, and its activation can lead to UA vasodilation involving the activation of the NO-cGMP pathway [72] and in a Ca^2+^ and extracellular-signal activated kinases (ERK1/2)-dependent manner [73]. Pregnant animals receiving ICI have been widely used to address nuclear ER-mediated mechanisms. However, this model is limited as it cannot exclude the role of GPR30-mediated estrogen signaling since ICI 182,780 is a high-affinity GPER1 agonist [74].

In summary, in keeping with our previous studies showing augmented UA H_2_S via selective UA EC and SM CBS expression in estrogen replacement treatment [26] and endogenous estrogens in ovine [29] and human pregnancy [28], our current study demonstrates, for the first time, that elevated endogenous estrogens stimulate UA EC and SM CBS expression with vascular bed-specific effects via an ER-dependent mechanism, further adding new evidence for the emerging role of enhanced UA H_2_S production as a new UA vasodilator to comprehend uterine hemodynamics regulation.

## 4. Materials and Methods

### 4.1. Chemicals and Antibodies

Monoclonal antibodies against CBS and CSE were obtained from Santa Cruz Biotechnology (Dallas, TX, USA). Anti-CD31 antibody was obtained from R&D Systems (Minneapolis, MN, USA). Prolong Gold antifade reagent with 4, 6-diamidino-2-phenylindole (DAPI), Alexa^488^ and Alexa^568^ conjugated goat anti-mouse immunoglobulin G (IgG) were obtained from Invitrogen (Carlsbad, CA, USA). Horseradish peroxidase-conjugated goat anti-mouse IgG was obtained from Cell Signaling (Beverly, MA, USA). ICI was from Tocris (Minneapolis, MN, USA). Bovine serum albumin (BSA) and all other chemicals, unless specified, were obtained from Sigma (St. Louis, MO, USA).

### 4.2. Animals and Treatments

Animal care and use procedures were in accordance with the National Institutes of Health guidelines (NIH Publication No. 85–23, revised 1996) with approval by the Institutional Animal Care and Use Committees at Shanghai Jiaotong University School of Medicine (A-2020-007), University of Wisconsin–Madison (V005847-R02), and University of California Irvine (AUP-21-156). Twelve-week-old Sprague Dawley pregnant [positive plug = gestation day (GD) 1] rats were randomly assigned to receive a subcutaneous injection (n = 9/group) of ICI (Tocris Cat #104711, 0.3 mg/rat in sesame oil) or vehicle (100 µL sesame oil) on GD19. Based on previous studies, this ICI dosage was chosen to effectively test its inhibitory effect on estrogen-induced gene expression in rodents in vivo [36,75]. These animals were purchased from Shanghai Silaike Experiment Animal Co., Ltd. and housed in an AAALAC-certified animal facility in the Center for Laboratory Animals at the Shanghai Jiaotong University School of Medicine, at 21 °C ± 1 °C with humidity of 55% ± 10%, and a 12 h light/12 h dark cycle with food and water ad libitum. Animals were sacrificed at 24 h post-injection to isolate UA, mesenteric arteries (MAs), and renal arteries (RAs). The arteries were snap-frozen immediately and then stored at −80 °C until analyzed. UA segments were also fixed in 4% paraformaldehyde for immunohistochemical analysis. Additional explant culture studies were performed with UA rings from nonpregnant (NP) and pregnant (P) GD20 Sprague-Dawley rats at the University of Wisconsin–Madison, as described previously [59]. All other analyses were performed at the University of California Irvine.

### 4.3. RNA Extraction, Reverse Transcription, and Quantitative Real-Time Polymerase Chain Reaction (qPCR)

RNA extraction, reverse transcription, and qPCR were performed with gene-specific primers listed in Table 1, as previously described [34,76]. Relative mRNA levels were quantified by using the comparative CT (ΔΔCt) method, with L19 as the internal reference control.

### 4.4. Immunofluorescence Microscopy and Image Analysis

Paraffin-embedded rat UA sections (5 µm) were deparaffinized in xylene and rehydrated. Antigen retrieval was achieved via boiling in 10 mM sodium citrate buffer for 15 min. Autofluorescence was quenched using three 15 min washes with 300 mM glycine in phosphate-buffered saline (PBS) at room temperature (RMT). After blocking non-specific binding in 1% BSA-PBS at RMT for 30 min, the sections were incubated with 1 μg/mL anti-CD31 in 0.5% BSA/PBS overnight at 4 °C. Following three 5 min washes in PBS, the sections were incubated with Alexa^568^ mouse IgG (2 μg/mL) at RMT for 1 h. After three 20 min washes in PBS, sections were blocked with 1% BSA/PBS and then incubated with 1 μg/mL of anti-CBS or anti-CSE antibodies overnight at 4 °C, followed by Alexa^488^ rabbit IgG or Alexa^488^ mouse IgG (2 μg/mL) at RMT for 1 h. IgG was used as a negative control. The sections were washed and mounted with SlowFade gold antifade mount containing DAPI (Invitrogen) to label the cell nuclei. The sections were examined under a confocal laser scanning microscope Olympus FV3000 (Olympus Corporation, Tokyo, Japan). Images were acquired to quantify the levels of CBS and CSE proteins in EC and SMC, as previously described [26,34].

### 4.5. Statistical Analysis

Data are presented as means ± SEM and analyzed via one-way analysis of variance (ANOVA), followed by the Bonferroni test for multiple comparisons using *SigmaStat*14 (Systat Software Inc, Palo Alto, CA, USA). Student’s *t*-test was used to compare NP vs. P groups. *p* < 0.05 was considered statistically significant unless indicated in the figure legends.

## Figures and Tables

**Figure 1 ijms-24-14384-f001:**
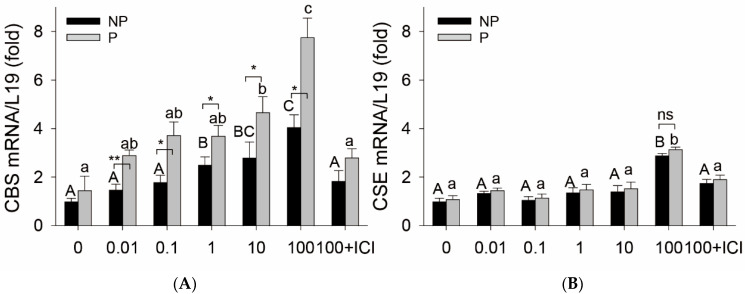
Effects of estradiol-17β on CBS and CSE expression in isolated uterine arteries. Endothelium-intact uterine artery (UA) rings from nonpregnant (NP) and pregnant (P, day 20) rats were treated with estradiol-17β (E2β, 0.01–100 nM) for 24 h. Total RNA was extracted to measure mRNAs of cystathionine β-synthase (CBS) and cystathionine γ-lyase (CSE) using quantitative real-time PCR (qPCR) using gene-specific primers listed in Table 1; L19 was measured as an internal control for quantitation. Data (means ± SEM) were summarized from 3 different rats. Bars with different superscripts differ significantly, *p* < 0.05 vs. untreated controls. * *p* < 0.05, ** *p* < 0.01, NP vs. P rats; ns: not significant.

**Figure 2 ijms-24-14384-f002:**
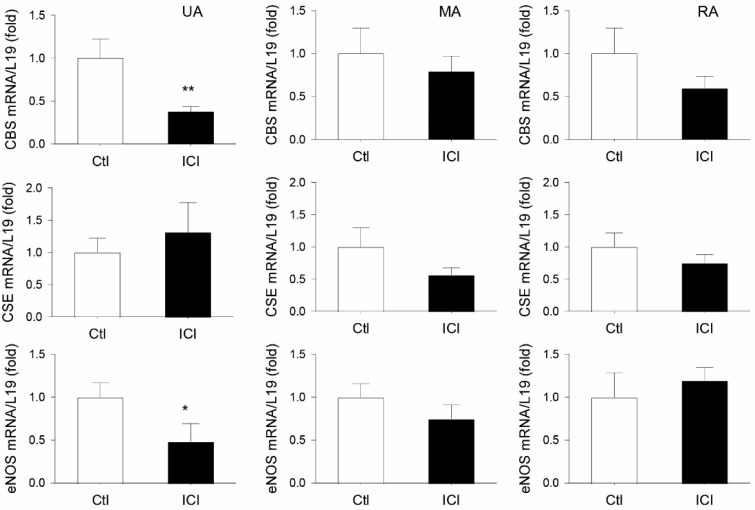
Effects of ICI 182, 760 on uterine and systemic (mesenteric and renal) artery CBS, CSE, and eNOS mRNA expression in pregnant rats in vivo. Time pregnant rats on gestation day 19 were treated with either sesame oil alone (Ctl) or with a specific estrogen receptor (ER) antagonist ICI 182, 780 (ICI, 0.3 mg/rat). Rats (n = 8) were sacrificed at 24 h after injection. Uterine (UA), mesenteric (MA), and renal (RA) arteries were collected to analyze mRNAs of cystathionine β-synthase (CBS), cystathionine γ-lyase (CSE), and endothelial nitric oxide synthase (eNOS) via qPCR with gene-specific primers listed in Table 1; L19 mRNA was measured as an internal control for quantitation. Data (means ± SEM) were summarized from artery samples of 5 different rats/group. * *p* < 0.05, ** *p* < 0.01 vs. vehicle (Ctl) treated controls.

**Figure 3 ijms-24-14384-f003:**
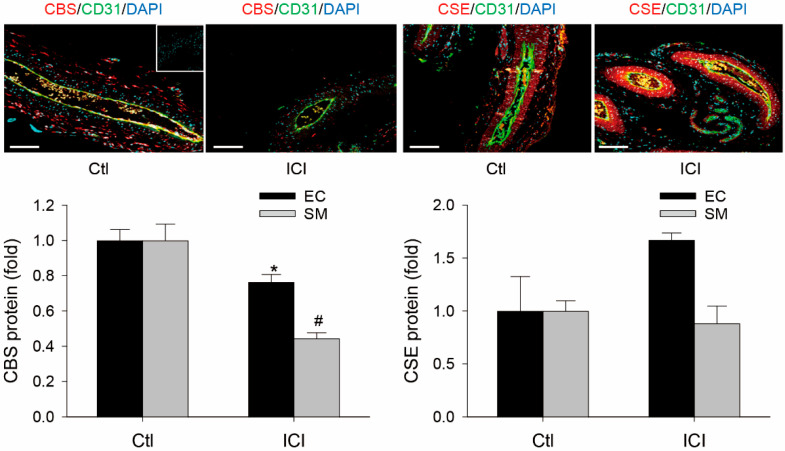
Effects of ICI 182, 760 on uterine artery CBS and CSE protein expression in pregnant rats in vivo. Uterine arteries (UAs) were collected from pregnant (gestation day 20) rats at 24 h treatment with vehicle (Ctl) or ICI 182, 780 (ICI, 0.3 mg/rat). Paraffin-embedded UA sections (5 µM) were subjected to immunofluorescence labeling of cystathionine β-synthase (CBS) and cystathionine γ-lyase (CSE) proteins using specific CBS or CSE antibodies, with CD31 antibody for co-labeling endothelial cells (ECs) distinct from smooth muscle cells (SMCs). After incubation with corresponding fluorescently labeled secondary antibodies, sections were mounted with DAPI to label nuclei and examined under confocal microscopy. IgG was used as negative control (insert). Images were taken to determine CBS and CSE proteins (relative green fluorescence intensity; RFI) using Image J and summarized as fold changes relative to untreated smooth muscles. Data (means ± SEM) were summarized from UA sections from three different rats. * and #, *p* < 0.05 vs. vehicle (Ctl) treated. Scale bar = 100 μm.

**Table 1 ijms-24-14384-t001:** Primers used for RT-qPCR.

Gene	Forward	Reverse	Product Size
CBS	TGAGATTGTGAGGACGCCCAC	TCGCACTGCTGCAGGATCTC	177 bp
CSE	AGCGATCACACCACAGACCAAG	ATCAGCACCCAGAGCCAAAGG	178 bp
eNOS	TACAGAGCAGCAAATCCAC	CAGGCTGCAGTCCTTTGAT	813 bp
L19	GGACCCCAATGAAACCAACG	GTGTTCTTCTAGCATCGAGC	129 bp

## Data Availability

Original data presented in the study are included in the article. Further inquiries can be directed to the corresponding author.

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
