# Peer review of "ICI 182,780 Attenuates Selective Upregulation of Uterine Artery Cystathionine β-Synthase Expression in Rat Pregnancy"

_ijms, 2023, doi:10.3390/ijms241814384_

Round 1
Reviewer 1 Report
The authors use a pregnant rat model treated with ICI to test the hypothesis that endogenous estrogens stimulate UA CBS expression via specific ER in vivo. Previous studies from this research group have demonstrated that the enhanced UA production of H2S/CBS, the third member of the gasotransmitter family, plays a key role in regulation of the increased uterine blood flow during pregnancy. The current results further demonstrate that elevated endogenous estrogens stimulate UA EC and SM CBS expression with vascular bed-specific effects via ER-dependent mechanism. These findings provide new evidence for an emerging role of enhanced UA production of H2S as a new UA vasodilator to comprehend uterine hemodynamics regulation.
Following concerns need to be address:
1) please correct the typo error: “1Dong-bao Chen and PhD”.
2) Please rewrite the sentence: “Maternal vascular adaptations in pregnancy result in gestation age dependent up to 20-50-fold increases in uterine blood flow (UtBF) for the delivery of maternal nutrients and oxygen to the fetus and the exhaust of respiratory gases and wastes from the fetus [3, 4].”
3) Please provide a rationale for treatment with E2β (0.01 to 100 nM). Are these concentrations of E2β in the physiological range?
4) Please discuss the selectivity of ICI on ER and potential side effect.
5) Previous studies seem demonstrate a high expression level of CBS/H2S in PUA as compared to NPUA, however, this study showing no difference in CBS mRNA between NP and PUA at baseline (Figure 1), but significant increase in CBS mRNA after E2 stimulation. Authors may need to further discuss the possibility to cause this difference. The rationale to use L19 as an internal control for mRNA quantitation is unclear, which may make the difference as compared to previous studies.
6) Pleas shorten the Discussion Section, need to more focus on the current findings and how your findings contributing to the field, instead of taking too much space to discuss previous studies/findings in the field.
Author Response
Responses to reviewer 1
1) please correct the typo error: “1Dong-bao Chen and PhD”.
Response: Corrected as needed.
2) Please rewrite the sentence: “Maternal vascular adaptations in pregnancy result in gestation age dependent up to 20-50-fold increases in uterine blood flow (UtBF) for the delivery of maternal nutrients and oxygen to the fetus and the exhaust of respiratory gases and wastes from the fetus [3, 4].”
Response: Modified as “Maternal vascular adaptations to pregnancy result in gestation age-dependent up to 20-50-fold increases in uterine blood flow (UtBF), mandatory for delivering maternal nutrients and oxygen to the fetus and for exhausting respiratory gases and wastes from the fetus [3, 4].
3) Please provide a rationale for treatment with E2β (0.01 to 100 nM). Are these concentrations of E2β in the physiological range?
Response: the E2β concentrations 0.01 to 100 nM tested are physiological (0.01 to 1 nM) and supraphysiological concentrations (10-100 nM) E2β, which can be seen as physiological concentrations in pregnancy, especially considering other estrogen species are also involved. We added more discussion to clarify the physiological relevance to these concentrations tested.
4) Please discuss the selectivity of ICI on ER and potential side effect.
Response: ICI is considered as a “pure” antiestrogen that can work on both ERα and ERβ with high affinity. However, in some studies, ICI can activate the membrane GPR30 that mediates estrogen resposnes through activating intracellular kinase pathways. We discussed this effect of ICI as a limitation of the ICI model.
5) Previous studies seem demonstrate a high expression level of CBS/H2S in PUA as compared to NPUA, however, this study showing no difference in CBS mRNA between NP and PUA at baseline (Figure 1), but significant increase in CBS mRNA after E2 stimulation. Authors may need to further discuss the possibility to cause this difference. The rationale to use L19 as an internal control for mRNA quantitation is unclear, which may make the difference as compared to previous studies.
Response: We used L-19 as an internal reference for qPCR studies. We expanded discussion to this result.
6) Pleas shorten the Discussion Section, need to more focus on the current findings and how your findings contributing to the field, instead of taking too much space to discuss previous studies/findings in the field.
Response: We agree with the comments on “more focus on the current findings” and added more interpretations of the data and limitations to accommodate. However, I hope that you can agree that citing the historic literature does help the readers to follow this important topic.

Reviewer 2 Report
In this manuscript by Bai et al. entitled “ICI 182,780 attenuates selective upregulation of uterine artery CBS expression in rat pregnancy” the authors showed that in vivo reduction of estrogen receptor (ER) activity decreases the expression of cystathionine b-synthase (CBS) in pregnant rat uterine arteries (UA). UA vasodilation is an important mechanism to increase blood flow during pregnancy allowing a proper maternal-fetal interchange of nutrients and gases. The present work shows that expression of CBS is reduced in pregnant UA by the in vivo treatment with a selective ER antagonist (ICI 182,780), this reduction was specific for uterine circulation as it was not observed in mesenteric or renal arteries. These results highlight an important mechanism by which estrogens promote H2S production in UA, suggesting an alternative mechanism, besides NO production, for UA vasodilation during pregnancy. The manuscript's main strength is the use of an in vivo antagonist treatment to determine the role of endogenous estrogen signaling. The paper is well presented, and the results support the conclusions. A few minor edits would increase this manuscript’s impact.
Comments:
1) The dose of ICI 182,780 given to the pregnant rats is not clear. The authors described the dose in the Abstract and the legends of Figures 2 and 3 as 0.3 µg/rat. However, in the Methods section is listed as 0.3 mg/rat. Furthermore, in the Discussion is noted as 0.3 µg/kg/rat, which is the correct way to describe a dose, per body weight. Please correct these discrepancies and use the appropriate units to describe drug dosage.
2) No rationale was provided for the dosage of ICI 182,780 utilized, is this dose comparable to the concentrations used in the ex vivo studies (Figure 1)? Please include the rationale for using this dosage.
3) The second sentence in the Introduction is not clear, consider rephrasing.
4) Figure 2 legend says “monetary” but should say “mesentery.”
5) Figure legends, please include “(Ctl)” after “sesame oil alone” in legend of Figure 2 and after "vehicle" in legend of Figure 3.
6) Figure 3 legend states “UA were collected from pregnant (gestation day 19) rats” but in the text is described that these were GD20 rats (which is the same GD as the mRNA expression studies). Please correct this discrepancy.
7) There is a typo in the Abstract: “lase” should be “lyase.”
Author Response
Responses to reviewer 2
The manuscript's main strength is the use of an in vivo antagonist treatment to determine the role of endogenous estrogen signaling. The paper is well presented, and the results support the conclusions. A few minor edits would increase this manuscript’s impact.
Response: We thank you very much for this comment and corrected all minor edits as suggested.
1) The dose of ICI 182,780 given to the pregnant rats is not clear. The authors described the dose in the Abstract and the legends of Figures 2 and 3 as 0.3 µg/rat. However, in the Methods section is listed as 0.3 mg/rat. Furthermore, in the Discussion is noted as 0.3 µg/kg/rat, which is the correct way to describe a dose, per body weight. Please correct these discrepancies and use the appropriate units to describe drug dosage.
Response: Thank you very much for pointing out this mistake. We used 0.3 mg/rat in 100 µl sesame oil in the study, which was administrated via subcutaneous injection. This has been corrected.
2) No rationale was provided for the dosage of ICI 182,780 utilized, is this dose comparable to the concentrations used in the ex vivo studies (Figure 1)? Please include the rationale for using this dosage.
Response: A range of 1-20 mg/day/rat ICI doses have been used in many previous studies in rodents in vivo, and pharmacological studies even tested high doses. It is difficult to compare in vitro and in vivo studies; however, ICI is considered as a pure antiestrogen and 1 uM dose was chosen as at this dose ICI exhibits maximum inhibitory effects on E2β-induced resposnes, with IC50 of approximately 2 × 10−7 m. We choose 0.3 mg/rat based on previous studies, which have been added in method.
3) The second sentence in the Introduction is not clear, consider rephrasing.
Response: Modified as “Maternal vascular adaptations to pregnancy result in gestation age-dependent up to 20-50-fold increases in uterine blood flow (UtBF), mandatory for the delivery of maternal nutrients and oxygen to the fetus and for the exhaust of respiratory gases and wastes from the fetus [3, 4].
4) Figure 2 legend says “monetary” but should say “mesentery.”
Response: Changed as needed. Thank you.
5) Figure legends, please include “(Ctl)” after “sesame oil alone” in legend of Figure 2 and after "vehicle" in legend of Figure 3.
Response: Changed as needed. Thank you.
6) Figure 3 legend states “UA were collected from pregnant (gestation day 19) rats” but in the text is described that these were GD20 rats (which is the same GD as the mRNA expression studies). Please correct this discrepancy.
Response: The rats were treated on GD19 and collected on GD20, changed as needed. Thank you.
7) There is a typo in the Abstract: “lase” should be “lyase.”
Response: Changed as needed. Thank you.